# Modeling Pneumonia-Induced Bloodstream Infection Using Graph Theory to Estimate Hospital Mortality

**Dimitrios Zikos [1],* and Maria Athanasopoulou [2]**

[1]   College of Health Professions, Central Michigan University, Mount Pleasant, MI 48859, USA
[2]   Languages Department, CUNY Queens College, New York, NY 11367, USA;
     maria.athanasopoulou@qc.cuny.edu
*   Correspondence: zikos1d@cmich.edu; Tel.: +1-989-774-1589

**Abstract:** Hospital-acquired pneumonia (HAP) bloodstream infections comprise a major cause of crude hospital mortality. This is a cross-sectional study that used claims data from the Centers for Medicare and Medicaid Services ($N$ = 565,875). The study objective is to represent the progression of pneumonia-induced bloodstream infections using graph theory principles, where each path of the graph represents a different scenario of bloodstream-infection progression, and aims to further estimate the likelihood if hospital death for each path. To disseminate the results, the study makes available a prototype applet to navigate various paths of the graph interactively. Bayesian probabilities were calculated for each scenario, and multivariate logistic regression was conducted to estimate the adjusted OR for inpatient death after controlling for patient age, sex, and comorbidities. The mortality rate ranged from 4.99% for patients admitted with community pneumonia without bloodstream infection and reached 63.18% for cases admitted with bloodstream infection that progressed to hospital septicemia, sepsis, and septic shock. The prototype applet can be used to unfold bloodstream infection events and their associated risk for mortality and could be used in university curricula to assist educators in helping students understand the progression of pneumonia-induced bloodstream infections in a data-driven way.

**Keywords:** health informatics; bloodstream infection; hospital-acquired pneumonia; event recognition; graph theory

## 1. Introduction

Hospital-acquired pneumonia (HAP) refers to any pneumonia contracted by a patient in a hospital at least 48–72 h after being admitted [1]. It is an infection that was not present before the patient came to the hospital and therefore is distinguished from community-acquired pneumonia. HAP is the second most common hospital-acquired infection. It is the most common cause of death among hospital-acquired infections [2] and is usually caused by a bacterial infection. Barber et al., in their review study, recognized HAP as a major cause of deaths, notably in patients with severe underlying conditions, and suggested that the development of new diagnostic tools and therapeutic methods was urgent to face the epidemic of antibiotic-resistant pathogens [3].

Factors that increase the risk of HAP include weakened cough, mechanical ventilation, tracheostomy tubes, suctioning and weakened immune system from disease or medications [2]. To diagnose HAP, there must be evidence on the chest x-ray as well as fever, elevated white blood cell count, or the presence of secretions from the lungs. Hospitalized patients with HAP receive antibiotic therapy, while other lung medications may be given to help loosen and remove thick secretions from the lungs. Oxygen may be provided to maintain good oxygen levels. Some of the components for HAP

prevention include clean hands, use antibacterial hand gel before and after each patient interaction, and the use of gloves during direct patient contact.

For some patients, HAP often progresses to septicemia, then severe sepsis or even septic shock. Early recognition of these patients might help in reducing morbidity and mortality. According to de Lange et al., elevated systemic levels of pro-inflammatory cytokines at the time of diagnosis of hospital-acquired pneumonia appear to be indicative of subsequent progression to septic shock [4]. HAP is different from community pneumonia since it is very often caused by methicillin-resistant staphylococcus aureus (MRSA).

The progression of a disease can be described as a temporal sequence of diagnosis events. In the case of pneumonia, the disease may, unfortunately, oftentimes progress to septicemia. Septicemia is a bacterial infection that spreads into the bloodstream [5]. Severe sepsis, that could be the next phase of a worsening septicemia, is the body's response to that infection, during which the immune system will trigger extreme and potentially dangerous, whole-body inflammation [6]. Sepsis, in turn, when it causes dangerously low blood pressure (shock), is called septic shock.

Almirante et al. studied the predictors of mortality among patients with Candida bloodstream infection is Spain and observed a very high (44%) mortality rate for patients with candidemia [7]. There is a scarcity of studies that examine differences in the mortality rate between hospital and community-onset bloodstream infections. One of these studies was conducted by Diekama et al., who examined the epidemiology and outcome of nosocomial and community-onset bloodstream infections, and found that bloodstream infections cause substantial morbidity and mortality. According to this study, the crude mortality for community-onset infections was 14%, but was much higher, at 34%, for nosocomial infections [8]. Other researchers have shown that a bloodstream infection has a severe impact on critically ill patients. To determine the excess mortality attributable to hospital-acquired bloodstream infections, Smith et al. found that among critically ill patients with bloodstream infections, observed mortality significantly exceeded the predicted value. Critically ill patients who develop nosocomial bloodstream infections are at greater risk of death than patients with comparable severity of illness without this complication [9]. These results are similar to those of a previous study that examined hospital mortality among patients in a Surgical intensive care unit (SICU). According to that study, nosocomial bloodstream infection complicated 2.67 per 100 admissions to the SICU and the crude mortality rates for a patient who developed a bloodstream infection was 50% and only 15% for patients who did not [10]. There are a few studies that study HAP-specific bloodstream infection and its association with crude mortality. Sopena et al. studied, in a prospective design, non-ICU HAP and found that the mean incidence of HAP was 3 ± 1.4 cases per 1000 hospital admissions, and mortality was 26%, with 13.9% being attributed to pneumonia [11].

While the aforementioned studies examined the impact of bloodstream infections on hospital outcomes of care, including inpatient mortality, none of them, to the authors' knowledge has examined the progression of bloodstream infections and how each phase of a pneumonia-induced bloodstream infection is associated with an increase to the likelihood of hospital death. Existing studies follow a traditional statistical design approach and were not designed to study, in a data science manner, the bloodstream infection as a progressive clinical event of multiple steps.

Since the progression of a bloodstream infection can be described as a temporal sequence of diagnoses (septicemia → severe sepsis → septic shock), this sequence makes up a clinical event. Event recognition is the detection of events that are considered relevant for processing [12]. Examples of event recognition consist of recognizing human activities on video content [13] and extracting interesting stories from social media. In healthcare, there are diverse event recognition paradigms, such as the identification of cardiac arrhythmia events [14], and the spread of an epidemic [15]. At the hospital, the association of those attributes that may form sequential clinical events have been explained in previous studies [16]. The knowledge of interesting clinical events can benefit medical education, quality assurance hospital committees who study high-risk inpatient scenarios, and clinicians who are informed about scenarios that are likely to unfold during a patient encounter.

In acknowledgment of this opportunity, we designed a retrospective cross-sectional study to represent the progression of pneumonia-induced bloodstream infections using graph theory principles, where each path of the graph is a different scenario of bloodstream-infection progression, and we furthermore estimated the likelihood for inpatient death for each path. To disseminate results more effectively, we also developed a prototype applet to navigate various pneumonia-induced bloodstream infection paths of the graph, interactively. The study contribution is summarized below: (a) A new approach is introduced for the study of the effect of pneumonia-induced bloodstream infections on hospital mortality. In this approach, the representation of the progression is based on directed graphs. (b) The attributable risk for inpatient death is quantified for each step of the progression by examining each path as a predictor using multivariate analysis. (c) A prototype applet is made available, allowing the user to navigate various paths of the graph and learn about the risk for inpatient mortality. This applet can be used as a data-driven tool in health curricula. Figure 1 shows the contextual framework of the study, with three different types of pneumonia-induced bloodstream infection events: Those with community onset and progression, those with hospital onset and progression, and mixed events with community onset and hospital progression.

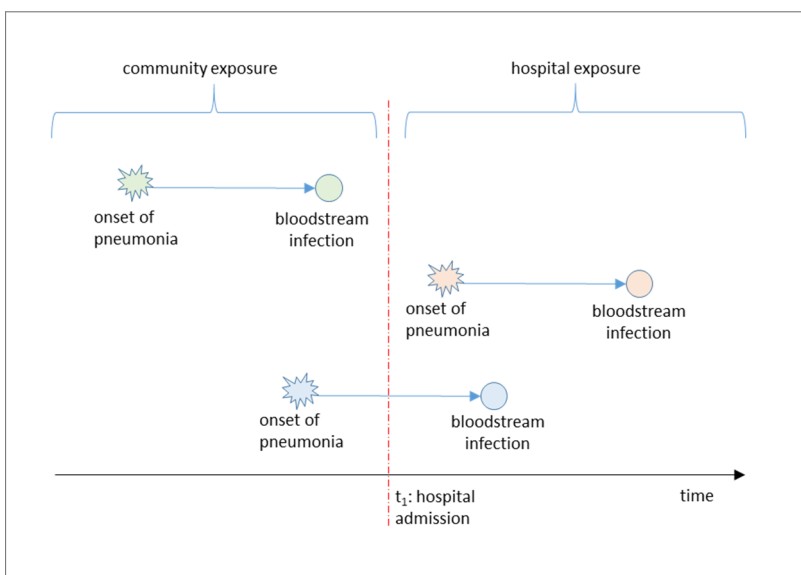

**Figure 1.** The contextual framework of the study.

## 2. Materials and Methods

### 2.1. Data Sources

This research used a large Limited Data Set (LDS) from the Centers for Medicare and Medicaid Services (CMS) with 565,875 records, each representing an inpatient Medicare case [17]. The file contains information about patient admission, patient demographics, the diagnoses in International Classification of Diseases format (principal and secondary), hospital-acquired conditions, medical procedures, disposition information, hospital charges, and service use variables. This dataset was used to extract information about the presence of the following attributes: Pneumonia, Septicemia, Severe Sepsis, and Septic Shock. For each of these four diagnoses, a 'Present of Admission' (POA) indicator was used to differentiate them as a hospital-, or non-hospital-acquired. Additionally, information about the following comorbidities was extracted: Cancer, Diabetes, Acute Myocardial Infarction (AMI), Chronic Obstructive Pulmonary Disease (COPD), Liver Disease, Asthma, Diabetes, and Hypertension, as well as the age group and gender for each patient. Finally, the outcome of interest, hospital death, was extracted as a dichotomous variable 'died'.

## *2.2. Approach*

Eight variables were used as nodes to estimate the clinical scenarios of bloodstream infections: Pneumonia(c), Septicemia(c), Severe Sepsis(c), Septic Shock(c), Pneumonia(h), Septicemia(h), Severe Sepsis(h), and Septic Shock(h). The (c) notation represents the appearance of the condition before the admission to the hospital (community exposure); the (h) notation represents a hospital-acquired exposure. Each of these eight variables is represented on the dataset as a dichotomous feature-node (0/1). The eight nodes can altogether be represented as a directed graph (Figure 2). The graph can be navigated for every possible path of pneumonia-induced bloodstream infections. Two medical domain constraints were used to construct this graph: (a) the mandatory sequentiality of the bloodstream infection: a pneumonia always precedes septicemia, septicemia always precedes severe sepsis, while severe sepsis always precedes septic shock. (b) Community exposure always precedes hospital exposure. Table 1 shows the four phases of the bloodstream infection progression, represented as graph nodes, their degree, and associated child nodes.

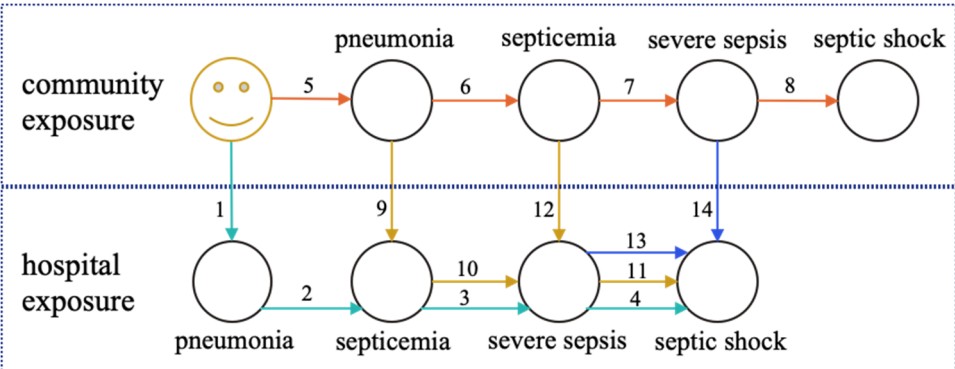

**Figure 2.** Directed graph of pneumonia-induced bloodstream infection progression.

**Table 1.** Graph nodes, their out-degree, and associated child nodes.

| Graph Node | Degree | Child Node(s) |
|---|---|---|
| Pneumonia(c) (root note) | 2 | Septicemia(c), Septicemia(h) |
| Septicemia(c) | 2 | SevereSepsis(c), SevereSepsis(h) |
| SevereSepsis(c) | 2 | SepticShock(p), SepticShock(h) |
| SepticShock(c) | 0 | none (leaf node) |
| Pneumonia(h) (root node) | 1 | Septicemia(h) |
| Septicemia(h) | 1 | SevereSepsis(h) |
| SevereSepsis(h) | 1 | SepticShock(h) |
| SepticShock(h) | 0 | none (leaf node) |

Because of the aforementioned domain constraints, there are fourteen different paths, each one representing a different bloodstream infection progression scenario. Each path is composed of edges as numbered in Figure 2. Table 2 summarizes the event that each path represents, as well as the condition to estimate the probability for hospital death, for each path.

For each one of these paths, the probability of inpatient mortality was calculated. For each patient, the database includes information for the following dichotomous features: {Pneumonia(c), Septicemia(c), SevereSepsis(c), SepticShock(c), Pneumonia(h), Septicemia(h), SevereSepsis(h), SepticShock(h), Death}. Each of these features were generated from the original CMS dataset, by combining the diagnostic codes with the 'present on admission' indicators. To calculate the conditional probabilities, the database was queried with SQL queries. The result of each query (each representing one of the aforementioned paths) was stored onto a new 'results' table.

For instance, to calculate the probability for path 1 (Figure 2), the following logic was used:

```
SELECT AVG(Died) FROM CMS_TABLE WHERE
Pneumonia(c)=0 AND Septicemia(c)=0 AND SevereSepsis(c)=0 AND Shock(c)=0 AND Pneumonia(h)=1;
```

**Table 2.** Paths of pneumonia-induced bloodstream infections.

| Path | Patient Was Admitted To The Hospital | Probability Of Inpatient Death GIVEN: |
|------|--------------------------------------|----------------------------------------|
| 1 | without community-onset bloodstream infection but developed HAP | `AND(Pneumonia(c)=N, Septicemia(c)=N, SevereSepsis(c)=N, Shock(c)=N, Pneumonia(h)=Y)` |
| 1, 2 | without community-onset bloodstream infection but developed HAP that progressed to septicemia | `AND(Pneumonia(c)=N, Septicemia(c)=N, SevereSepsis(c)=N, Shock(c)'=N, Pneumonia(h)=Y Septicemia(h)=Y)` |
| 1, 2, 3 | without community-onset bloodstream infection but developed HAP which progressed to septicemia & severe sepsis | `AND(Pneumonia(c)=N, Septicemia(c)=N, SevereSepsis(c)=N, Shock(c)=N, Pneumonia(h)=Y, Septicemia(h)=Y, SevereSepsis(h)=Y)` |
| 1, 2, 3, 4 | without community-onset bloodstream infection but developed HAP that progressed to septicemia, severe sepsis & shock | `AND(Pneumonia(c)=N, Septicemia(c)=N, SevereSepsis(c)=N, Shock(c)=N, Pneumonia(h)=Y, Septicemia(h)=Y, SevereSepsis(h)=Y, Shock(h)=Y)` |
| 5 | with community-onset community pneumonia | `AND(Pneumonia(c)=Y, Septicemia(c)=N, SevereSepsis(c)=N, Shock(c)=N)` |
| 5, 6 | with community-onset pneumonia & septicemia | `AND(Pneumonia(c)=Y, Septicemia(c)=Y, SevereSepsis(c)=N, Shock(c)=N)` |
| 5, 6, 7 | with community-onset pneumonia, septicemia & severe sepsis | `AND(Pneumonia(c)=Y, Septicemia(c)=Y, SevereSepsis(c)=Y, Shock(c)=N)` |
| 5, 6, 7, 8 | with community-onset pneumonia, septicemia, severe sepsis & shock | `AND(Pneumonia(c)=Y, Septicemia(c)=Y, SevereSepsis(c)=Y, Shock(c)=Y)` |
| 5, 9 | with community-onset pneumonia which progressed to septicemia in-hospital | `AND(Pneumonia(c)=Y, Septicemia(c)=N, SevereSepsis(c)=N, Shock(c)=N, Septicemia(h)=Y)` |
| 5, 9, 3 | with community-onset pneumonia which progressed to septicemia & severe sepsis in-hospital | `AND(Pneumonia(c)=Y, Septicemia(c)=N, SevereSepsis(c)=N, Shock(c)=N, Septicemia(h)=Y, SevereSepsis(h)=Y)` |
| 5, 9, 3, 4 | with community-onset pneumonia which progressed to septicemia, severe sepsis and shock in-hospital | `AND(Pneumonia(c)=Y, Septicemia(c)=N, SevereSepsis(c)=N, Shock(c)=N, Septicemia(h)=Y, SevereSepsis(h)=Y, Shock(h)=Y)` |
| 5, 6, 10 | with community-onset pneumonia & septicemia, which progressed to severe sepsis in-hospital | `AND(Pneumonia(p)=Y, Septicemia(p)=Y, SevereSepsis(p)=N, Shock(p)=N, SevereSepsis(h)=Y)` |
| 5, 6, 10, 4 | with community-onset pneumonia & septicemia, progressing to severe sepsis & shock in-hospital | `AND(Pneumonia(p)=Y, Septicemia(p)=Y, SevereSepsis(p)=N, Shock(p)=N, SevereSepsis(h)=Y, Shock(h)=Y)` |
| 5, 6, 7, 11 | with a community-onset pneumonia septicemia & severe sepsis which progressed to shock in-hospital | `AND(Pneumonia(p)=Y, Septicemia(p)=Y, SevereSepsis(p)=Y, Shock(p)=N, Shock(h)=Y)` |

Due to the dichotomous nature of the 'died' feature, the result of this query is a fraction that takes value between 0 and 1, representing the conditional probability of death for the query condition. Additionally, each path was represented as a composite variable and inserted into multiple binary logistic regression models to examine the likelihood of each scenario for hospital death after adjusting for patient age, sex, and comorbidities. These additional variables were available in the CMS dataset that this research used, and were inserted to the regression as control variables, for more accurate Odds Ratio (OR) estimates. These additional variables were not used in the path analysis. The analysis

tasks were completed with relational queries on Microsoft SQL server [18]. The regression analysis was conducted with Weka data mining software [19].

## 3. Results

### 3.1. Scenarios with Community Onset and Progression

There are four scenarios where the development of pneumonia or pneumonia-induced bloodstream infection happened before admission to the hospital. There are four paths that represent these scenarios: {5}, {5, 6}, {5, 6, 7}, {5, 6, 7, 8}. As Table 3 shows, the probability of hospital death is increased when a patient is admitted to the hospital when the pneumonia-induced bloodstream infection has worsened at the time of admission.

**Table 3.** Community onset of pneumonia induced bloodstream infection.

| Pneumonia Onset In The Community | Septicemia Onset In The Community | Severe Sepsis Onset In The Community | Septic Shock Onset In The Community | N | P(Death) |
|---|---|---|---|---|---|
| Yes | No | No | No | 36,294 | 4.99% |
| Yes | Yes | No | No | 5922 | 7.77% |
| Yes | Yes | Yes | No | 2400 | 17.04% |
| Yes | Yes | Yes | Yes | 2148 | 35.01% |

### 3.2. Scenarios with Hospital Onset and Progression

In four scenarios, the patient is admitted to the hospital pneumonia-free and develops pneumonia or pneumonia-induced bloodstream infection during the hospital stay. There are four paths that represent these events: {1}, {1, 2}, {1, 2, 3}, {1, 2, 3, 4}. The probability of death when the pneumonia-induced bloodstream infection has an in-hospital onset is significantly higher than for community infection (Table 4). Characteristically, when a patient is admitted to the hospital with pneumonia-induced septic shock (Scenario 1), the probability for death is 35.01%. If septic shock is developed in-hospital as a result of HAP (Scenario 2), the probability of death is increased to 52.57%.

**Table 4.** Hospital onset of pneumonia-induced bloodstream infection.

| Pneumonia Onset In-Hospital | Septicemia Onset In-Hospital | Severe Sepsis Onset In-Hospital | Septic Shock Onset In-Hospital | N | P(Death) |
|---|---|---|---|---|---|
| Yes | Not yet known | Not yet known | Not yet known | 4193 | 13.71% |
| Yes | Yes | Not yet known | Not yet known | 677 | 33.97% |
| Yes | Yes | Yes | Not yet known | 389 | 45.75% |
| Yes | Yes | Yes | Yes | 253 | 52.57% |

### 3.3. Mixed Scenarios with Community Onset and Further Progression in Hospital

In the last six scenarios, pneumonia or a pneumonia-induced bloodstream infection preexisted but was further progressed to more severe stages, in-hospital. There are six paths that represent such events: {5, 9}, {5, 9, 3}, {5, 3, 4}, {5, 6, 10}, {5, 6, 10, 4}, {5, 6, 7, 11}. These mixed scenarios of community pneumonia that progressed to nosocomial bloodstream infection, demonstrated the highest mortality rates. As Table 5 shows, the crude mortality rate of community pneumonias without bloodstream infection that progressed to septic shock in hospital is 63.18% (Table 5).

The infographic (Figure 3) visualizes the mortality rate for each of the 14 scenarios. In green in represented community (pre-hospitalization) onset of bloodstream infection phases, while the nosocomial ones are in red. It is apparent that the most lethal combination of pneumonia-induced bloodstream infection is one that started as community pneumonia, and progressed to septicemia, severe sepsis, and finally septic shock, in hospital.

**Table 5.** Community onset of pneumonia-induced bloodstream infection with hospital progression.

| Before Hospitalization | | | | During Hospitalization | | | | N | P(Death) |
|---|---|---|---|---|---|---|---|---|---|
| Pneumonia | Septicemia | Severe Sepsis | Septic Shock | Pneumonia | Septicemia | Severe Sepsis | Septic Shock | | |
| Yes | No | No | No | No | Yes | Yet unknown | Yet unknown | 485 | 41.44% |
| Yes | No | No | No | No | Yes | Yes | Yet unknown | 295 | 54.23% |
| Yes | No | No | No | No | Yes | Yes | Yes | 182 | 63.18% |
| Yes | Yes | No | No | No | No | Yes | Yet unknown | 51 | 41.17% |
| Yes | Yes | No | No | No | No | Yes | Yes | 35 | 48.37% |
| Yes | Yes | Yes | No | No | No | No | Yes | 90 | 53.33% |

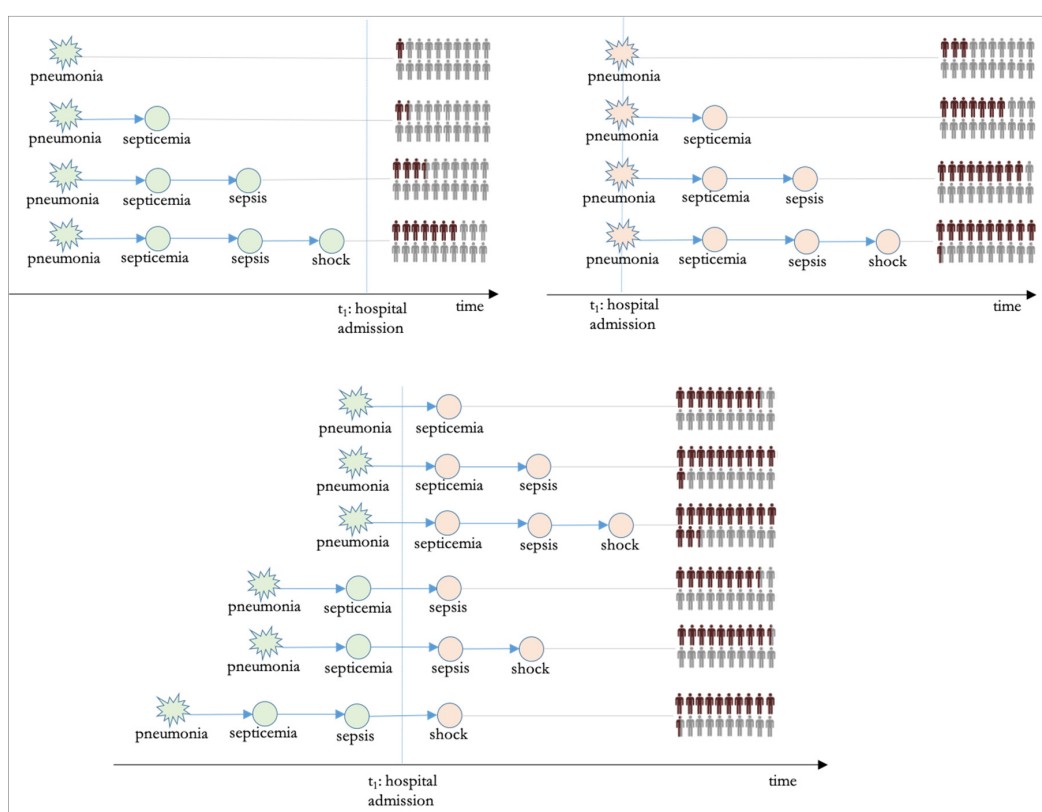

**Figure 3.** Infographic of the proportion of deaths for each pneumonia-induced bloodstream infection scenario.

*3.4. Regression Analysis*

To examine the likelihood of hospital death for each bloodstream infection scenario, after controlling for patient demographics and comorbidities, multiple logistic regression analysis was conducted, with the dependent variable being the dichotomous variable 'died'. This is a separate, supplementary investigation of the outcome of hospital mortality and is not connected with the graph estimation and the bloodstream progression navigation. The independent variables were inserted into the model using the Enter method. Fourteen regression models were created. In each model, each of the fourteen scenarios was inserted into the model, represented as a composite independent variable. For example, the scenario *"Admitted without a community-onset bloodstream infection & developed nosocomial pneumonia"* was represented as a composite variable that takes value '1' only when the following condition is true: *AND(Pneumonia(c) = 0, Septicemia(c) = 0, Sepsis(c) = 0, Shock(c) = 0, Pneumonia(h) = 1).* Similarly, composite variables were created for all the remaining thirteen bloodstream infection scenarios. Table 6 shows the adjusted odds ratios (OR) for inpatient death for each of the bloodstream infection progression scenarios, after controlling for patient comorbidities, gender, and age. According to the results, the likelihood of inpatient death is the highest for the scenario of an admission with community pneumonia, which progresses in hospital to septicemia, severe sepsis and septic shock (OR = 40.86. 95% CI = 29.68–56.26). The presence of this composite scenario is associated with an

increase in the likelihood of inpatient death by 40 times. The likelihood of hospital death is higher in those scenarios where the bloodstream infection has worsened to a later stage (Table 6).

**Table 6.** Binary logistic regression to study the effect of bloodstream infection scenarios on hospital mortality.

| | Scenario (Composite Feature) | | | | | | | | Adjusted OR * | |
|---|---|---|---|---|---|---|---|---|---|---|
| BLR | Pn(c) x | Sept(c) x | Sepsis(c) x | Shock(c) x | Pn(h) x | Sept(h) x | Sepsis(h) x | Shock(h) | O.R | C.I |
| #1 | 1 | 0 | 0 | 0 | n/a | n/a | n/a | n/a | 1.37 | 1.30–1.44 |
| #2 | 1 | 1 | 0 | 0 | n/a | n/a | n/a | n/a | 2.06 | 1.86–2.27 |
| #3 | 1 | 1 | 1 | 0 | n/a | n/a | n/a | n/a | 4.52 | 4.03–5.07 |
| #4 | 1 | 1 | 1 | 1 | n/a | n/a | n/a | n/a | 11.91 | 10.81–13.13 |
| #5 | 0 | 0 | 0 | 0 | 1 | n/a | n/a | n/a | 4.11 | 3.74–4.51 |
| #6 | 0 | 0 | 0 | 0 | 1 | 1 | n/a | n/a | 13.48 | 11.36–16.01 |
| #7 | 0 | 0 | 0 | 0 | 1 | 1 | 1 | n/a | 22.17 | 17.87–27.50 |
| #8 | 0 | 0 | 0 | 0 | 1 | 1 | 1 | 1 | 29.16 | 22.36–38.03 |
| #9 | 1 | 0 | 0 | 0 | 0 | 1 | n/a | n/a | 17.61 | 14.52–21.37 |
| #10 | 1 | 0 | 0 | 0 | 0 | 1 | 1 | n/a | 29.21 | 22.87–37.31 |
| #11 | 1 | 0 | 0 | 0 | 0 | 1 | 1 | 1 | 40.86 | 29.68–56.26 |
| #12 | 1 | 1 | 0 | 0 | 0 | 0 | 1 | n/a | 16.11 | 8.85–29.31 |
| #13 | 1 | 1 | 0 | 0 | 0 | 0 | 1 | 1 | 19.46 | 9.57–39.57 |
| #14 | 1 | 1 | 1 | 0 | 0 | 0 | 0 | 1 | 24.53 | 15.69–38.34 |

* adjusted for patient comorbidities, age, and sex.

### 3.5. The IPN (Infection Progression Navigator) Applet

The IPN applet is an interactive graph navigator where the user can develop scenarios of pneumonia-induced bloodstream infections. This prototype applet shows the user the risk for hospital death, as well as the attributable risk for death, for each bloodstream infection phase they add to their scenario. The applet was developed with Microsoft Visual Studio 2017 and the data were loaded on a relational database, using Microsoft SQL Server 2017 [18]. The more recent version of the IPN navigator can be downloaded here: https://tinyurl.com/yd9wt9lf.

The applet allows the user to create scenarios of a bloodstream infection progression, step by step. When the user opens the applet, he/she is welcomed by a digital event creator canvas. As shown in Figure 4, the user has the freedom of choice to develop any of the possible combinations of the pneumonia induced bloodstream infection, as described in this study. When the user clicks the 'add' button for the next progression step (either pre-admission or post-admission) the system shows three pieces of information:

(i)     The risk for hospital death for the user selected scenario;
(ii)    The attributable risk of the last user-selected addition on the risk for hospital death; and
(iii)   The likelihood for the infection to progress to the step that the user selected.

The user can open more than one snapshots of an in-progress scenario in two separate windows, for easier overview and comparisons. The applet maintains data-application independence: It is easy to extend the functionality so that it will not be limited to the experiments and the paths that were explained on this paper (pneumonia-induced infections). In that respect, the backend database of the applet can be updated, at any time, with new data on bloodstream infections induced by other bacterial diseases and maintain the same frontend functionality.

*IPN* was used as a supplementary educational tool in the framework of the undergraduate university course *'Health Care Quality Improvements'*, which is part of a health administration major, to demonstrate to students the importance of preventive measures to reduce the incidence of hospital infections, and the impact that such efforts would have on patient outcomes. Twenty-four students participated in a class meeting and were presented with a summary of this research and a live demonstration of the application in action. Then they were then asked to unfold their own scenarios

and to discuss the attributable risk for inpatient death during the progression of pneumonia-induced bloodstream infections. Students also examined differences in the risk between scenarios of community and hospital exposure.

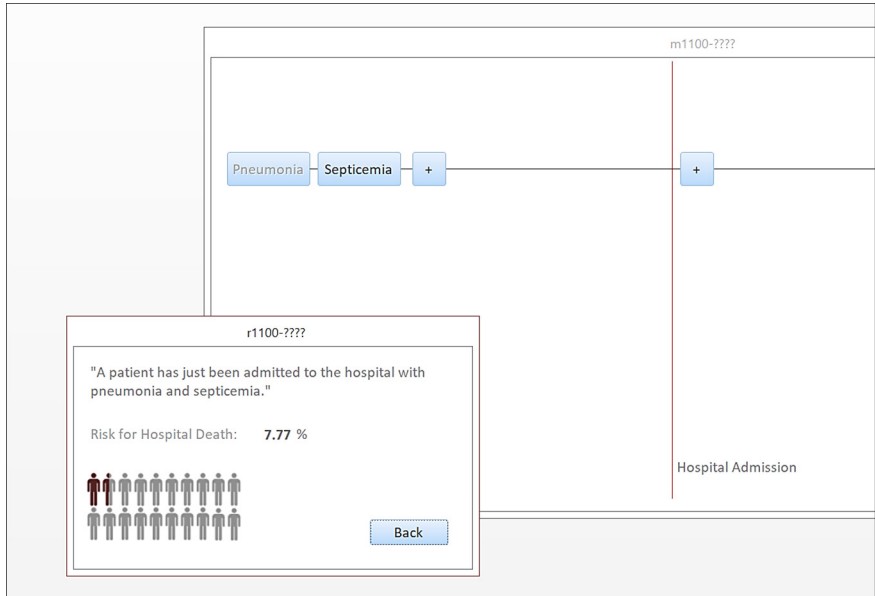

**Figure 4.** An instance of the INP applet. The user initiated a scenario, of a community pneumonia which progressed to septicemia before the hospital-admission, to see the risk for hospital death.

## 4. Discussion

This study used a novel representation of pneumonia-induced bloodstream infections, based on graph theory. The study used a dataset with real medical claims data from the Centers for Medicare and Medicaid Services. The scenarios that the research examined were based on the established clinical knowledge that bloodstream infection progression is sequential and ordered, with septicemia always preceding sepsis, which always precedes septic shock, with many articles examining this progression [20]. Each stage of the infection was represented as a graph node, and the graph paths were therefore different scenarios of bloodstream infections. The probability of death was found to be lower when a patient was admitted to the hospital with community pneumonia that had not progressed to a bloodstream infection. At that time point, and considering that it is yet unknown whether the patient will develop a nosocomial infection or not, the mortality is only 4.99%. Patients with community pneumonia typically respond well to antibiotics, which they usually receive early on, soon after admission to the hospital. Apparently, this is a prophylaxis from further progression to a life-threatening bloodstream infection. On the other hand, when pneumonia is HAP, hospital mortality is much higher, at 13.71%. The main reason for this finding is that these cases of pneumonia are most likely to be methicillin resistant. Additionally, when a HAP-induced bloodstream infection develops in the hospital, the crude mortality is much higher compared to pneumonia-induced bloodstream infections imported from the community (45.75% vs. 17.04%, for the progression up the stage of severe sepsis). Septicemia and sepsis have been extensively studied in the literature as risk factors for hospital death. The Centers for Disease Control and Prevention (CDC) make available US State-level statistics of septicemia mortality [21], and it is evident that septicemia is a major risk factor. In the present study, we have furthermore shown that the likelihood for death is alarmingly high when septicemia is induced by pneumonia and further progresses to severe sepsis.

The mixed scenarios are of particular interest. These are scenarios where pneumonia began in the community, but the bloodstream infection progressed in-hospital. Some of these scenarios were found to have the highest rates of crude hospital mortality, at up to 63%. Those very high mortality

rates can be explained by considering that a patient whose—already known—pneumonia cannot be treated and progresses to a life-threatening bloodstream infection is a critical patient who has a high risk of dying because of the underlying case mix of his/her conditions. Another explanation is that the pneumonia brought from the community may not be easy to treat because of its being caused by antibiotic-resistant microorganisms, despite those being atypical in the community.

The results of our study confirm the knowledge that HAPs are more lethal than community cases of pneumonia [8,22]. Additionally, the study confirms the knowledge that a pneumonia-induced bloodstream infection is associated with very high crude mortality rates [11]. What this research adds to existing knowledge is the breakdown of the risk for each phase of a bloodstream infection. The progression in the various scenarios was represented in the form of clinical events with temporal sequences that can be displayed on computer graphs. The authors believe that approaches similar to the proposed 14-path directed graph can be further used in future research to study bloodstream infections as hospital events and examine the attributable risk for each progression phase.

The primary intended audience for the prototype navigator applet consists of university students and their instructors in health science programs, such as nursing and health administration. In such programs, students discuss the significance of preventive measures in lowering the risk for hospital infections and its implications in hospital quality. Therefore, navigating interactive graphs to create scenarios of bloodstream infections and discussing the risk in class can enhance data-driven aspects of college education.

The authors would finally like to discuss two limitations of this work and strategies for future improvements. First of all, the dataset that the research used includes Medicare admissions in US hospitals. Medicare patients are, in their majority, >65 years old, and therefore the results should be interpreted cautiously for younger inpatient cases. Secondly, the research only examined pneumonia-induced bloodstream infections, and therefore the end-user has limited events to explore via the graph representation of events, a total of fourteen. Future work can extend the scope by examining other sources of bloodstream infections, such as UTIs and abscesses, to create a universal bloodstream infection navigator which covers more scenarios than the pneumonia-induced ones. Additionally, with more data records in hand, future work could make it be possible to explore bloodstream infection progression scenarios for specific patient demographics and age groups. This was not possible in the present work, because it would lead to over filtering the dataset and having an insufficient amount of records per path. Finally, the authors want to discuss that there is a non-perfect link between the node of pneumonia and that of septicemia. This is because a small portion of patients with pneumonia could have developed septicemia triggered by other acute conditions that coexist with pneumonia, and this inconsistency has not been quantified.

**Author Contributions:** D.Z. developed the target dataset for analysis, wrote the article, and supervised the research, M.A. provided recommendations about implications of findings, and wrote portions of the discussion. All authors have read and agreed to the published version of the manuscript.

**Funding:** This research is supported by the Faculty Research and Creative Endeavors (FRCE), grant number: 48530 (01/2019, $8,000), PI: Dimitrios Zikos, Grant Title: Training Platform for Computer-Assisted Construction and Risk Stratification of Comorbidity Profiles.

**Conflicts of Interest:** The authors declare no conflict of interest.

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
