# Peer review of "Modeling Pneumonia-Induced Bloodstream Infection Using Graph Theory to Estimate Hospital Mortality"

_technologies, doi:10.3390/technologies8020024_

Round 1
Reviewer 1 Report
The article proposes the use of graph theory to estimate hospital mortality. Particularly the authors focus on pneumonia-induced blood stream infection. The article is interesting and well written.
The authors introduce the medical problem for expert and not expert readers. The proposed approach is then explained and the results are discussed.
Both the theoretical and the experimental part are satisfactory.
In the following some minor issues to fix:
- figures: please use a different font to make the text more visible
- table 2: selected probabilities are 0 or 1. Is it reasonable to consider different values? Please discuss
- Paragraph 3.5: please better describe the applet and its functionalities. This is one point of originality of the article, but it has not been properly described. Please describe the screenshot in figure 4. It is not understandable.
Author Response
Thank you for taking the time to review our manuscript and provide constructive feedback! We have studied your critique carefully and provided responses to your recommendations, point by point, on the attached document.

Reviewer 2 Report
The paper presents a tool to categorise the progression of pneumonia cases using a simple graph structure in which different paths from a start node (exposure) to an end node (eg septic shock) represent different clinical evolutions. The background data and general idea are rather straightforward and clearly explained. I believe the tool can be useful in the context in which the authors have developed it, although it appears to produce an output that focuses on the bare essential in terms of data and variables. I would recommend publication if the authors can address the issues listed below.
Major: the article gives no details about how the probabilities of each path and all other quantities are computed from data. Details about the whole analysis (including regression) should be added to the main text, especially as the study makes use of variables that are not accounted for in the description of paths (implying that, at least in principle, much more information can be extracted from it)
Major: L239-247 discuss the interpretation of the results. While I understand this is not the main point of this work, I wonder whether the scenarios discussed are realistic. Can the authors point to literature that supports their conclusions in some way? The lack of references in this part is not ideal.
Major: I did not quite understand from the text whether the applet is accessible. If not, do the authors plan on making it accessible somehow? I understand that background data are sensible but I don’t see how this technology can be useful if it’s not usable, perhaps with a dummy dataset that can be used as a guide by people interested in using it.
Major: Can this method lead to a simple/usable predictive (ie not only descriptive) tool? For instance, given the values of some clinical variables, one could train a neural network to compute the probability of certain paths when the patients are in the early stages of the progression. I think that would be a potentially interesting extension.
Minor: The “degree” to which the authors refer in Table 1 is usually called “out-degree” (to distinguish it from the number of incoming links)
Author Response

(The authors gave the same response as above.)

Round 2
Reviewer 2 Report
Most of my concerns were addressed